# Involvement of Both Extrinsic and Intrinsic Apoptotic Pathways in Tridecylpyrrolidine-Diol Derivative-Induced Apoptosis In Vitro

**DOI:** 10.3390/ijms241411696

**Published:** 2023-07-20

**Authors:** Natalia Nosalova, Alexandra Keselakova, Martin Kello, Miroslava Martinkova, Dominika Fabianova, Martina Bago Pilatova

**Affiliations:** 1Department of Pharmacology, Faculty of Medicine, P.J. Šafárik University, 040 01 Košice, Slovakia; natalia.nosalova@student.upjs.sk (N.N.); alexandra.macejova@student.upjs.sk (A.K.); 2Department of Organic Chemistry, Faculty of Science, Institute of Chemical Sciences, P.J. Šafárik University, 040 01 Košice, Slovakia; miroslava.martinkova@upjs.sk (M.M.); dominika.jackova@gmail.com (D.F.)

**Keywords:** pyrrolidine, apoptosis, migration, colorectal cancer

## Abstract

Despite the decreasing trend in mortality from colorectal cancer, this disease still remains the third most common cause of death from cancer. In the present study, we investigated the antiproliferative and pro-apoptotic effects of (2*S*,3*S*,4*R*)-2-tridecylpyrrolidine-3,4-diol hydrochloride on colon cancer cells (Caco-2 and HCT116). The antiproliferative effect and IC_50_ values were determined by the MTT and BrdU assays. Flow cytometry, qRT-PCR and Western blot were used to study the cellular and molecular mechanisms involved in the induction of apoptotic pathways. Colon cancer cell migration was monitored by the scratch assay. Concentration-dependent cytotoxic and antiproliferative effects on both cell lines, with IC_50_ values of 3.2 ± 0.1 μmol/L (MTT) vs. 6.46 ± 2.84 μmol/L (BrdU) for HCT116 and 2.17 ± 1.5 μmol/L (MTT) vs. 1.59 ± 0.72 μmol/L (BrdU), for Caco-2 were observed. The results showed that tridecylpyrrolidine-induced apoptosis was associated with the externalization of phosphatidylserine, reduced mitochondrial membrane potential (MMP) accompanied by the activation of casp-3/7, the cleavage of PARP and casp-8, the overexpression of TNF-α and FasL and the dysregulation of Bcl-2 family proteins. Inhibition of the migration of treated cells across the wound area was detected. Taken together, our data show that the anticancer effects of tridecylpyrrolidine analogues in colon cancer cells are mediated by antiproliferative activity, the induction of both extrinsic and intrinsic apoptotic pathways and the inhibition of cell migration.

## 1. Introduction

Colorectal carcinoma (CRC) is one of the main causes of cancer-related mortality worldwide [1]. According to Globocan statistics, approximately 1.93 million new cases are diagnosed yearly [2]. Patients with unresectable CRC are treated with cytotoxic drugs (5-fluorouracil, oxaliplatin, irinotecan) or targeted biological agents (anti-VEGF—bevacizumab, ramucirumab; or anti-EGFR—cetuximab, panitumumab) [3,4,5]. Cancers are often associated with therapy resistance and drug toxicity. For these reasons, it is necessary to continuously identify new molecules with effective antitumor activity, low toxicity and minimal side effects.

Recent studies have suggested that pyrrolidines belong to the studied molecules with potential antiproliferative effects. Pyrrolidines can be described as organic compounds biosynthesized in numerous plant species, such as *Piper nigrum* [6], *Cannabis sativa* [7] and *Pandanus amaryllifolius* [8]. Chemically, pyrrolidines are tetrahydropyrroles or cyclic amines whose five-membered ring consists of four carbon atoms and one nitrogen atom (Figure 1A) [9]. Pyrrolidine alkaloids have promising biological effects. A recent review [10] provides an overview of the basic effects of pyrrolidine alkaloids, including antibacterial [11], neurological [12] and cytotoxic [13]. One of the most important neuroactive pyrrolidine alkaloids is nicotine [14], but habbemines A and B, which bind to opioid receptors, are also noteworthy [12]. (2*R*,3*S*,4*R*,5*R*) pyrrolidine-(1-hydroxyethyl)-3,4-diol, isolated from the sea sponge *Haliclona* sp. in the Red Sea, and Aegyptolidines A and B, from the fungus *Aspergillus aegyptiacus* [15], are known for their cytotoxic activity against several tumor cell lines [16]. The cytotoxic effects of pyrrolidine derivatives are exerted through various mechanisms, including cell cycle arrest [17,18], the induction of apoptosis and the modulation of various signaling pathways, among others [19,20]. Tumor cells exhibit high migration rates, leading to invasion and metastasis. Interestingly, pyrrolidine derivatives have shown anti-migratory activity that can reduce the severity and aggressiveness of cancer [18]. 

Apoptosis, the most studied among the programmed cell deaths, is a caspase-dependent process regulated by executioner and regulatory molecules. There are two main apoptotic pathways, which can influence each other, and there is considerable crosstalk between them [21,22]. The extrinsic death receptor pathway involves transmembrane receptor-mediated interactions. The stimulus for its initiation is specific death ligands binding to corresponding death receptors, such as TNF-α/TNFR1 and FasL/FasR, resulting in death-inducing signaling complex (DISC) formation and followed by initiator caspase-8 activation, which then activates the executioner caspase-3 [21,23]. The intrinsic mitochondrial pathway is regulated by proteins of the Bcl-2 family. Anti-apoptotic proteins from the Bcl-2 family include four domain proteins, Bcl-2, Bcl-xL, Mcl-1 and Bcl-w [22]. Proapoptotic BH3-only domain proteins can be subdivided into activators and sensitizers. Sensitizers Bik, Bmf, Bad, Noxa, Hrk and Bnip3 bind to anti-apoptotic Bcl-2 proteins, resulting in the release of activators. Activators Bid, Bim and Puma are able to bind to proapoptotic multidomain proteins Bax and Bak, leading to their direct activation and oligomerization [21,24]. The insertion of Bax/Bak into mitochondrial membranes induces mitochondrial outer membrane permeabilization, followed by the release of pro-apoptotic proteins such as cytochrome c from the mitochondria into the cytosol. Cytochrome c combines with Apaf-1 and procaspase-9 to form a multi-protein complex, the apoptosome, which participates in the activation of caspase-9 and subsequently the executioner caspase-3′s activation [21,23,25]. Both pathways proceed to the final apoptotic execution phase, controlling the breakdown of cells and leading to cell death [26]. Based on published data, this study aimed to investigate the anticancer potential of the newly synthesized pyrrolidine derivative SS13, focusing on the antiproliferative and pro-apoptotic effects of the tested compound on an in vitro model of colorectal cancer. A set of experiments was performed to better understand the molecular mechanisms of SS13-induced apoptosis.

## 2. Results

### 2.1. Effect of SS13 on Cell Viability and Proliferation

Colorimetric MTT and BrdU proliferation assays were used to test the effects of SS13 on the proliferation of HCT116 and Caco-2. MTT assay data showed that the tested compound had a concentration-dependent cytotoxic and antiproliferative effect on both cell lines (Figure 2). The IC_50_ values of 3.2 ± 0.1 μmol/L for HCT116 and 2.17 ± 1.5 μmol/L for Caco-2 after 72 h were obtained from the MTT reduction assay, based on measuring the metabolic activity of cells (Table 1). Furthermore, the MTT test was used to evaluate the effect of pyrrolidine derivative SS13 on the cell viability of epithelial non-cancerous control cell line MCF-10A (Figure 2C); as shown in Table 1, the calculated selectivity index for HCT116 was 2.75, and it was 4 for Caco-2 cells. Moreover, we performed the MTT assay with DMSO-treated HCT116, Caco-2 and MCF-10A cells. As seen in Figure 2, DMSO showed, in all tested cell lines, minimal or no toxicity/inhibition of metabolic activity. The comparison of SS13 and cisplatin’s IC_50_ values in both cancer cells showed higher cytotoxicity for SS13 than cisplatin (Table 1). In addition, the selectivity index for Cis-Pt was similar to that of SS13 in the HCT116 cell line, but the Caco-2 cell line had no selectivity for Cis-Pt. Moreover, the BrdU proliferation assay showed similar results, with an IC_50_ 6.46 ± 2.84 μmol/L for HCT116 and 1.59 ± 0.72 μmol/L for Caco-2 (Figure 2D,E). Both screening tests were performed simultaneously. The MTT assay demonstrates the metabolic activity of cells. The BrdU assay is a more sensitive tool to investigate the antiproliferative potential of SS13, because this method is based on incorporating 5-bromo-2′-deoxyuridine (BrdU) into DNA during cell replication, thereby determining pure cell proliferation. Based on the obtained results, we decided to use an SS13 concentration of 7 μmol/L for HCT116 and 2.5 μmol/L for Caco-2 in the subsequent analyses, since Caco-2 cells showed higher sensitivity to the tested compound than HCT116 cells (Figure 2).

### 2.2. AO/PI Fluorescent Staining

Changes in membrane permeabilization, which could be a sign of ongoing cell damage and cell death, induced by pyrrolidine SS13 were analyzed with double AO/PI staining. AO and PI are fluorescent dyes that can bind nucleic acids. AO can diffuse into cells with intact membrane integrity and PI to cells with a disrupted membrane. With this method, the cell population can be divided into living (green), apoptotic (yellow/orange) and dead (red) cells. The incubation of HCT116 and Caco-2 cells with pyrrolidine derivative SS13 led, in a time-dependent manner, to a significant decrease in the number of living cells and a significant increase in the number of possible apoptotic and dead cells with partially or completely disturbed membranes, leading to increased membrane permeabilization (Figure 3). 

### 2.3. Scratch Assay 

The scratch/wound healing assay was performed to monitor the effect of SS13 on the cell migration of HCT116 and Caco-2 cells. As seen in Figure 4A,B, the cells not exposed to SS13 repopulated the wounded area in a time-dependent manner. In all groups treated with SS13 at the IC_10_ (1.4 µM of HCT116 and 0.5 µM of Caco-2), IC_20_ (2.8 µM of HCT116 and 1 µM of Caco-2) and IC_50_ (7 µM of HCT116 and 2.5 µM of Caco-2), we noticed significantly reduced migration and wound-healing properties in seeded cells. A significant decrease in the number of cells compared to the untreated cells, in a dose- and time-dependent manner (Figure 4C,D), was also observed. We suppose that pyrrolidine derivative SS13 can influence the migration of colorectal cancer cells at sublethal concentrations. In the IC_50_ groups, the massive detachment of the cells as the result of ongoing apoptotic changes was observed. 

### 2.4. Cell Cycle Analysis 

Flow cytometric analysis was performed to identify whether the tested compound induced cell cycle arrest. We noticed G1 cell cycle arrest after 24 h in the HCT116 cell line, but in Caco-2 cells, we did not observe this phenomenon (Table 2, Figure 5). In addition, the significant time-dependent accumulation of events (fragmented nuclei) in the sub-G0 subpopulation was observed, which is a typical marker of apoptosis. 

### 2.5. Annexin V-FITC/PI Staining

The process of apoptosis is associated with changes in the structure of the cell membrane, which leads to the externalization of phosphatidylserine, the main component of the lipid bilayer. Annexin V specifically interacts with phosphatidylserine on the cell surface, and PI is a DNA intercalating agent able to diffuse across membranes with disturbed integrity. Double staining with Annexin V and PI is used to distinguish living cells (An−/PI−), cells in the early apoptotic stage (An+/PI−) and cells in the late apoptotic stage (An+/PI+) or necrotic cells (An−/PI+). As shown in Figure 6, pyrrolidine derivative SS13 induced significant changes in both cell lines. A significant increase was observed in the early apoptotic cell population after 24 h of treatment and in cells in the late stage of apoptosis after 48 h and 72 h of treatment in the HCT116 cell line (Figure 6A,C). Compared to HCT116, Caco-2 cells showed an increase in the early phase of apoptosis after 24, 48 and 72 h of incubation with SS13 (Figure 6B,D).

### 2.6. Effect of SS13 on Intrinsic and Extrinsic Apoptotic Pathways

#### 2.6.1. Activation of Intrinsic Apoptotic Pathway

##### Determination of Mitochondrial Membrane Potential, Cytochrome c Release and Apaf-1

Apoptosis is a process that occurs through two main pathways, the intrinsic mitochondrial pathway and the extrinsic pathway mediated by death receptors. In the intrinsic pathway of apoptosis, the permeability of the outer mitochondrial membrane and the subsequent release of apoptogenic proteins into the cytosol increase. A key indicator of mitochondrial activity is the mitochondrial membrane potential. An MMP reduction is associated with mitochondrial damage and the release of cytochrome *c* from the intermembrane mitochondrial space [27]. We noticed a significant increase in the number of HCT116 (Figure 7A,C) and Caco-2 cells (Figure 7B,D) with reduced MMP in a time-dependent manner after SS13 treatment.

Furthermore, we observed the significant accumulation of cytosolic cytochrome *c* in both cell lines after all three exposure times with the tested compound (Figure 7E,F). Our results suggest that pyrrolidine-induced cell death is associated with mitochondrial damage and the release of cytochrome *c* into the cytosol. The release of cytochrome *c* triggers the oligomerization of apoptotic protease-activating factor-1 (Apaf-1), which plays an important role in the mitochondrial pathway of apoptosis [28]. SS13 treatment caused the overexpression of Apaf-1 in HCT116 after 72 h and in Caco-2 after 48 and 72 h, which can render cells more sensitive to apoptotic stimuli. 

##### Changes in Bcl Family

Bcl family proteins are regulators of the intracellular mechanism of apoptosis. RT-PCR analysis showed the significant upregulation of pro-apoptotic members of the Bcl-2 family. We observed a significant increase in the expression of the proapoptotic *BAX* gene after exposure to SS13 in both cell lines. The expression of proapoptotic *BIK* in the Caco-2 and HCT116 cell lines was different after incubation with SS13. In the HCT116 cell line, we first observed its downregulation after 24 and 48 h, with subsequent upregulation after 72 h. The opposite trend was noted in Caco-2 cells. First, its increased expression was observed after 24 and 48 h, followed by a significant decrease after 72 h (Table 3). Western blot analysis of the pro-apoptotic protein Bad showed an increase in total Bad in both cell lines (HCT116 only at 48 h). In the HCT116 cell line, we observed a simultaneous decrease in its phosphorylated form (pBad). Caco-2 cells did not show this phenomenon but, in contrast, we observed an increase in pBad (Figure 8). The analysis of anti-apoptotic *BCL2* and *BCL2L1* gene expression showed a significant reduction in both anti-apoptotic proteins in HCT116 cells after SS13 exposure (Table 3). The Western blot analysis corresponded with the data obtained by RT-PCR, and decreased levels of Bcl-xL and Bcl-2 as well as phosphorylated Bcl-2 (*p*-Bcl-2) were noted (Figure 8A). On the other hand, in Caco-2 cells, the significant upregulation of *BCL2* and *BCL2L1* expression was observed (Table 3), and the levels of Bcl-xL and p-Bcl-2 increased significantly when using the Western blot method (Figure 8B).

##### Activity of Caspase 3/7 and PARP Cleavage 

In response to mitochondrial damage and the release of cytochrome c, caspases can propagate the apoptotic process. Cysteine-aspartic proteases, caspases, can be activated in the initiation or execution phase of apoptosis [29]. Using this assay, we detected the percentage of live, apoptotic and dead cells based on caspase-3/7 activity in combination with the dead cell stain SYTOX™ AADvanced™. Figure 9 shows the SS13-activated caspase-3/7 in HCT116 and Caco-2 cells. We observed a significant increase in activated caspase-3/7 in HCT116, with the maximum increase observed after 24 h of SS13 treatment (Figure 9A), and in Caco-2, with a maximum after 48 h (Figure 9B) in apoptotic cells. In addition, a significant increase in the number of dead cells was noticed after 72 h treatment (Figure 9C,D). These activated proteases are responsible for the proteolytic cleavage of poly-(ADP-ribose)-polymerase (PARP). PARP is one of the repair enzymes involved in several cellular processes, such as replication or DNA repair [30]. The present study found a time-dependent increase in cleaved PARP after SS13 treatment in both HCT116 (Figure 9E) and Caco-2 cells (Figure 9F). 

##### Inhibitor of Apoptosis Protein (IAP) Family

X-linked inhibitor of apoptosis (XIAP) and survivin aurvivingvin are members of the IAP family, which inhibit the apoptotic process. XIAP and survivin have inhibitory effects on caspase activity. The gene expression of *XIAP* was significantly reduced at all monitored times (24, 48, 72 h) in HCT116 cells, but, in the Caco-2 cell line, *XIAP* was downregulated only after 72 h of treatment with the tested compound (Table 3). Our experiment, paradoxically, revealed a significant increase in *BIRC5* (survivin) expression in Caco-2 cells after 24 and 48 h of SS13 treatment and in HCT116 at all exposure times (Table 3).

#### 2.6.2. Activation of Extrinsic Apoptotic Pathway 

Pro-apoptotic ligands interacting with specialized cell surface death receptors activate the extrinsic pathway of apoptosis, also called the pathway of death receptors. FasL and TNF-α are pro-apoptotic/death ligands [31]. In our experiments, we noticed a significant increase in the expression of the pro-apoptotic gene *FASLG* after 24 and 48 h of SS13 treatment in Caco-2 cells and at all monitored time intervals (24, 48 and 72 h) in HCT116 cells. *TNF* gene expression in the HCT116 cell line showed significant upregulation only after 24 h of incubation with SS13, and, after 72 h, it was significantly downregulated. In Caco-2 cells, *TNF* expression increased at all three exposure times (Table 3). Cleavage of procaspase-8 into its active form is one of the hallmarks of the extrinsic apoptotic pathway. In our experiments, we observed the significant upregulation of *CASP8* expression after the 24 h incubation of HCT116 and Caco-2 with SS13 (Table 3), and the Western blot showed a significant increase in cleaved caspase-8 that peaked after 48 h of SS13 treatment in both cell lines (Figure 10). 

## 3. Discussion

Colorectal cancer is a global public health problem with high incidence and mortality worldwide. Nowadays, research on CRC is focused on developing more effective and less aggressive therapies [5]. Pyrrolidines, natural or synthetic substances, are interesting molecules with cytotoxic properties, and their antiproliferative activity has already been demonstrated in several studies. Modifying their chemical structure and synthesizing new derivatives could lead to the development of new substances with potential antitumor effects [32].

Our study examined the molecular mechanism of the cytotoxic effects induced by newly synthesized pyrrolidine derivative SS13 in HCT116 and Caco-2 colon cancer cells. We demonstrated that the tested compound significantly suppressed the proliferation of both tumor cell lines, with higher sensitivity in Caco-2, potentially because HCT116 is more resistant to many anticancer agents [32]. Some studies have shown that G1/S or G2/M phase cell cycle arrest can mediate pyrrolidine derivates’ cytotoxic effects [18,33]. Flow cytometric analysis was used to determine whether SS13 induced changes in the cell cycle. Our results showed that the tested pyrrolidine derivative caused G1 phase cell cycle arrest only after 24 h of treatment in HCT116 cells, but in the Caco-2 cell line, it did not induce any cell cycle arrest, as described in previous studies [18,33]; therefore, we hypothesized that the observed induction of apoptosis probably occurred through a different mechanism.

Apoptosis, a major mechanism of programmed cell death, may be triggered by intrinsic stimuli (oxidative stress, hypoxia) via the mitochondrial signaling pathway or by extrinsic stimuli through death receptors, such as Fas (CD95/APO1), tumor necrosis factor (TNF)-α or TNF-related apoptosis-inducing ligand (TRAIL) [34]. Apoptotic cell death results in chromatin condensation, DNA fragmentation, cell shrinkage and apoptotic body formation. Furthermore, mitochondrial membrane permeabilization, the externalization of phosphatidylserine and the activation of caspases are hallmarks of apoptosis [35]. Morphologically, AO/PI staining confirmed apoptosis, and the detection of PS externalization showed a decrease in the number of living cells and a significant increase in the number of apoptotic (early and late) and dead cells after SS13 exposure in colorectal carcinoma cells. These results were consistent with several studies in which time-dependent apoptosis was observed after treatment with pyrrolidine derivates [17,36]. The apoptotic process is associated with reduced MMP and the release of pro-apoptotic proteins, such as cytochrome *c*, to the cytosol. Cytochrome *c* plays a key role in the activation of the internal mitochondrial pathway of apoptosis. It forms an apoptosome with procaspase-9, and this proteolytically active complex subsequently triggers the activation of the caspase cascade [37]. The present study showed a significant decrease in MMP in a time-dependent manner, accompanied by the significant release of cytochrome *c* from mitochondria to cytosol and increased activity of effector caspases-3/7 in both HCT116 and Caco-2 cells after SS13 treatment. 

The enzyme PARP is a substrate for caspases involved in DNA repair. It is cleaved during apoptosis, which is responsible for inactivating PARP and subsequently preventing the repair of DNA damage [38]. In our experiments, increased caspase-3/7 activity contributed to the cleavage of PARP into 89 kDa fragments. Della Ragione and Chen described the similar degradation of PARP by effector caspases and the release of cytochrome *c* after pyrrolidine derivate treatment in cancer cells [19,27]. Bcl-2 family proteins play an important role in cell survival and cell death. These proteins are categorized as anti-apoptotic (Bcl-2, Bcl-xL) and pro-apoptotic, with only the BH3 domain (Bad, Bik), or executioner pro-apoptotic subfamilies (Bax, Bak). The first two groups of proteins influence the activation of executioner ones [39]. The balance between pro-apoptotic and anti-apoptotic proteins is important for cell homeostasis. The activation of executioner pro-apoptotic proteins can lead to the formation of pores in the outer mitochondrial membrane (MOM), resulting in a reduction in MMPs and subsequent apoptosis [40]. It is known that only non-phosphorylated Bad, which is translocated to the mitochondria, may displace Bax and dimerize with Bcl-xL and Bcl-2 and neutralize their anti-apoptotic effects. Executioner pro-apoptotic Bad is free and can initiate MOM permeability and cytochrome *c* release to cytosol. Phosphorylated Bad (p-Bad) cannot heterodimerize with Bcl-2 or Bcl-xL and is sequestered into the cytosol, preventing its pro-apoptotic activity on the mitochondrial membrane [41].

The total form of Bad increased significantly after treatment with the tested pyrrolidine derivative, and the follow-up Western blot analysis showed a significant reduction in p-Bad in HCT116 in a time-dependent manner. Although an increase in the level of p-Bad was found in Caco-2 cells after exposure to SS13, this mechanism is probably related to the self-protection of cancer cells. Our results showed the significant overexpression of pro-apoptotic *BAX* and *BIK* after SS13 treatment. The downregulation of anti-apoptotic *BCL2* and *BCL2L1* was detected in HCT116 cells. The data presented here are consistent with the findings of Morais and co-workers, who also demonstrated a decrease in the levels of anti-apoptotic proteins Bcl-2 and Bcl-xL in renal carcinoma cells [42]. The phosphorylation of Bcl-2 at Serine 70 suppresses its anti-apoptotic activity by inhibiting the interaction between Bcl-2 and Bax [43]. However, the p-Bcl-2 in HCT116 cells did not increase after SS13 treatment.

On the other hand, Caco-2 cells showed the significant upregulation of *BCL2* and *BCL2L1*, and Western blot analysis revealed an increase in the levels of Bcl-xL and p-Bcl-2 in Caco-2 cells after SS13 treatment. This can be explained by the fact that some stress kinases, such as JNK, can trigger the phosphorylation of the Bcl-2 protein [44]. An increase in anti-apoptotic proteins Bcl-xL and p-Bcl-2 could also be associated with another form of cell death, such as methuosis [45]. 

In addition, the antiproliferative effect of tested pyrrolidine derivative SS13 was associated with the induction of the extrinsic pathway of apoptosis. Death ligands, such as FasL, TNF-α and Apo2L/TRAIL, which interact with DR, a member of the tumor necrosis factor receptor (TNFR) superfamily, activate the pathway of death receptors (DR) [46]. The most important inducers of apoptosis include the activation of Fas, DR4 (TRAIL-R1) and DR5 (TRAIL-R2), which triggers the cell-extrinsic apoptotic cascade, followed by forming a death-inducing signaling complex (DISC). DISC often contains the pro-apoptotic protease procaspase-8 and the adaptor protein, Fas-associated protein with death domain (FADD) [47]. The DR-mediated activation of caspase-8 is either direct, by cleaving executioner caspases, or indirect, by inducting the intrinsic apoptotic pathway [48]. In the present study, gene expression analysis of death ligands showed a significant increase in the levels of *FASLG* expression in both cell lines.

Moreover, for the first time, the significant upregulation of *TNF* in both Caco-2 and HCT116 cells after SS13 treatment was observed in a time-dependent manner. The upregulation of *CASP8* expression and the significant cleavage of procaspase-8 to its active form were noted after exposure to pyrrolidine derivative SS13. These results suggest that SS13 activates the extrinsic pathway of apoptosis. Although these mechanisms of apoptosis induction after pyrrolidine derivative treatment in colorectal cancer cells have not been previously described, Song and co-workers documented the inhibitory and pro-apoptotic effect of pyrrolidine dithiocarbamate in the MCF-7 cell line *via* the caspase-8-mediated Fas pathway [49]. Suppressing the inhibitors of apoptosis proteins (IAP), such as XIAP, survivin or cIAP1, can also lead to the activation of the apoptotic machinery. There are eight known IAP proteins in humans, XIAP, cIAP1, cIAP2, ILP2, NAIP, BRUCE, Livin and survivin [50]. We demonstrated for the first time that pyrrolidine derivative SS13 decreased the gene expression of *XIAP*. On the other hand, the gene expression of *BIRC5* (survivin) increased significantly. However, XIAP and survivin are inhibitors of apoptosis; they represent multi-functional proteins that are involved in mitosis, autophagy and apoptosis regulation in cells. The positive regulator of survivin mRNA translation is the AKT/mTOR signaling pathway [51], which can regulate survivin’s function. 

One of the typical features of cancer cells is a high migratory rate. The migration of tumor cells from the primary site of growth and the invasiveness of cells can lead to the development of metastases. The tested SS13 showed an inhibitory effect on cell migration in a time-dependent manner, and these results are consistent with Omar and co-workers’ study, which demonstrated that some pyrrolidine derivatives could suppress the migratory activity of cancer cells [18].

## 4. Materials and Methods

### 4.1. Tested Compound

(2*S*,3*S*,4*R*)-2-tridecylpyrrolidine-3,4-diol hydrochloride (SS13, Figure 1) was synthesized by Fabiánová et al. (Department of Organic Chemistry, Institute of Chemical Sciences, Pavol Jozef Safarik University in Kosice). Pyrrolidine derivative SS13 was selected based on screening from a series of tested substances [52]. The studied compound was dissolved in dimethyl sulfoxide (DMSO; Sigma-Aldrich Chemie, Steinheim, Germany) with a final concentration of less than 0.02% DMSO in the cultured medium. This concentration exhibited no cytotoxicity toward cultured cells.

### 4.2. Cell Cultures

This study used two human colorectal cancer cell lines, HCT116 and Caco-2, and epithelial non-cancerous cells MCF-10A, obtained from the American Type Culture Collection (ATCC; Manassas, VA, USA). HCT116 cells were cultured in RPMI 1640 medium (Biosera, Kansas City, MO, USA) and Caco-2 in Dulbecco’s Modified Eagle’s Medium (DMEM). Both growth media were supplemented with antibiotic/antimycotic solution (Merck, Darmstadt, Germany) and 10% fetal bovine serum (FBS; Gibco, Thermo Scientific, Rockford, IL, USA). The MCF-10A cells were cultured in growth medium consisting of high-glucose Dulbecco’s Modified Eagle’s Medium F12 (DMEM F12; Biosera, Kansas City, MO, USA) supplemented with antibiotic/antimycotic solution, insulin (10 µg/mL final), EGF (final 20 ng/mL), hydrocortisone (0.5 µg/mL final)(all Merck, Darmstadt, Germany) and 10% fetal bovine serum. Cells were maintained in a humidified atmosphere containing 5% CO_2_ at 37 °C. Before all experiments, cell viability was greater than 95%. 

### 4.3. MTT Assay 

The cell viability of colorectal carcinoma cells HCT116, Caco-2 and epithelial non-cancerous MCF-10A was evaluated by the MTT assay, in which viable metabolically active cells reduce the MTT dye to purple formazan crystals [53]. Cells were seeded in 96-well culture plates (5 × 10^3^/well) and treated with the tested compound SS13 at different concentrations in the range of 1–15 µmol/L, as well as cisplatin (c = 1, 5, 10, 50, 100 µM) and DMSO (*v*/*v* 0.002, 0.01, 0.02, 0.1, 0.2%), in a final volume of 100 µL. Moreover, 0.02% DMSO is equivalent to a 10 μmol/L concentration of the tested substance (SS13). In our experiments, we used SS13 concentrations of 2.5 and 7 μmol/L, where the DMSO equivalent was 0.005 and 0.0014%, respectively. After 72 h incubation, 10 µL of MTT solution (5 mg/mL, Sigma-Aldrich Chemie, Steinheim, Germany) was added to each well and it was incubated at 37 °C in a 5% CO_2_ atmosphere for 4 h, and then the formazan precipitate was dissolved in 100 µL 10% sodium dodecyl sulfate (SDS). The absorbance was measured at a wavelength of 540 nm using the automated Cytation™ 3 Cell Imaging Multi-Mode Reader (Biotek, Winooski, VT, USA). The IC_50_ was determined as the concentration showing 50% cell growth inhibition compared to the control. The IC_50_ values were calculated as a predictive TREND model.

### 4.4. BrdU Proliferation Assay 

Cell proliferation was analyzed in parallel with the MTT assay and determined using a commercially available kit (Cell Proliferation ELISA, BrdU Colorimetric kit, Roche Diagnostics GmbH, Mannheim, Germany). HCT116 and Caco-2 cells were grown at 5 × 10^3^ cells/well in 96-well plates for 24 h and then treated with the studied compound (concentrations of 1, 2, 5, 10 and 15 µmol/L). After 48 h, the cells were labeled with BrdU labeling solution and incubated at 37 °C overnight. The cells were then fixed with FixDenat solution, incubated for 90 min with anti-BrdU peroxidase conjugate and washed three times with the BrdU wash solution. TMB substrate solution was added to the cells, and after 5 min of incubation, the reaction was stopped with 25 µL 1 M H_2_SO_4_. The blue color of the solution changed to yellow. The absorbance was measured at 450 nm by the automated Cytation^TM^ 3 Cell Imaging Multi-Mode Reader (Biotek, Winooski, VT, USA). The IC_50_ values were calculated based on the results of the BrdU proliferation assay. The IC_50_ values were calculated as a predictive TREND model.

### 4.5. AO/PI Staining

Apoptosis occurrence was analyzed by AO/PI staining using the fluorescence microscopy technique. AO and PI are fluorescent dyes that can bind nucleic acids. AO can diffuse into cells with intact membrane integrity and PI to cells with a disrupted membrane. With this method, the cell population can be divided into living (green), apoptotic (yellow/orange) and dead (red) cells. The cells were seeded in 6-well plates (1 × 10^5^ cells/well) and incubated for 24, 48 and 72 h with tested compound SS13. After incubation, the cell lines HCT116 and Caco-2 were washed with PBS, fixed with 4% paraformaldehyde for 20 min and washed again with PBS. An acridine orange ((AO)/propidium iodide (PI) stock solution was prepared using acridine orange and propidium iodide (Sigma Aldrich, Steinheim, Germany) St. Louis, MO, USA) at a final concentration of 10 µg/mL each. The staining solution was added to the wells, and the cells were incubated for 1 h in the dark at room temperature. Cells washed with PBS in the plates were dried and detected using an automated Cytation™ 3 Cell Imaging Multi-Mode Reader (Biotek, Winooski, VT, USA; excitation filters 360/40 nm, 485/20 nm; emitting filters 460/40 nm, 528/20 nm, 620/40 nm).

### 4.6. Scratch Assay

The migratory potential of cells was analyzed by the scratch assay as follows. The HCT116 (1 ×10^6^ cells/well) and Caco-2 (6.5 × 10^5^ cells/well) cells were seeded in 6-well plates and attached to the surface under standard incubation conditions for 24 h. The following day, monolayers in the centers of the wells were scratched in a straight line using an SPL Scar™ scratcher (SPL Life Science, Pocheon, Republic of Korea). The cells were then carefully washed with PBS and treated with SS13 at different concentrations (IC_10_, IC_20_, IC_50_). At 0, 24, 48 and 72 h post-wounding, images were captured with the Cytation™ 3 Cell Imaging Multi-Mode Reader (Biotek, Winooski, VT, USA). Before each recording of the wounding area, the cells were stained with Hoechst 33,342 (Sigma Aldrich, St. Louis, MO, USA). The Gene 5 software (Biotek, Winooski, VT, USA) was used to determine the wounded area changes by cell count analysis. 

### 4.7. Flow Cytometric Analysis

HCT116 and Caco-2 cells were seeded in Petri dishes and treated with the tested compound for 24, 48 and 72 h. After incubation, the cells were harvested and centrifuged. The pellet was washed and resuspended in PBS and divided for analyses. The BD FACSCalibur flow cytometer (Becton-Dickinson, San Jose, CA, USA) was used to measure fluorescence. 

#### 4.7.1. Annexin V/PI Staining 

Early and late apoptosis is accompanied by the externalization of phosphatidylserine, which was analyzed by Annexin V/PI staining. The pellet was resuspended in 100 µL of PBS and stained with a solution of Annexin V–Alexa Fluor^®^ 647 (1:300, Thermo Scientific, Rockford, IL, USA) for 15 min of incubation in the dark and washed in PBS, and, subsequently, 1 µL of PI (0.025 mg/mL) (Sigma Aldrich, St. Louis, MO, USA) was added to the samples. The FL-2 (585/42) vs. FL-4 (661/16) channels were used to analyze the data. A minimum of 1 × 10^4^ events were analyzed per sample, and all experiments were performed in triplicate. FC data were analyzed with the FlowJo software v.10 (BD Biosciences, San Jose, CA, USA).

#### 4.7.2. Cell Cycle Analysis

Cell cycle changes were analyzed via the incorporation of PI into DNA and measured by flow cytometry. The following four subpopulations were visualized: sub-G0 (dead and apoptotic with fragmented nuclei), G1, S and G2 phases. The cultured cells were harvested, washed with PBS and fixed in cold 70% ethanol. The samples were stored at −20 °C for at least 24 h. The cells were washed in PBS, followed by incubation in a staining solution for 30 min at room temperature in the dark. The composition of the staining solution was 0.5 mg/mL ribonuclease A, a 0.2% final concentration of Triton X-100 and 0.025 mg/mL PI in 500 µL of PBS (all from Sigma Aldrich, St. Louis, MO, USA). The FL-2 (585/42) channel was used to collect the fluorescence signal. A minimum of 1 × 10^4^ events were analyzed per sample, and all experiments were performed in triplicate. The cell aggregates were distinguished using FL-2-W vs. FL-2-A plots. The histograms and cell population markers were obtained using the FlowJo software v.10 (BD Biosciences, San Jose, CA, USA), using the Dean–Jett–Fox model for all samples.

#### 4.7.3. Detection of Mitochondrial Membrane Potential Changes 

Mitochondrial membrane potential (MMP) was indicated by the retention of the red-orange positively charged dye tetramethylrhodamine ethyl ester (TMRE; Molecular Probes, Eugene, OR, USA). TMRE (Ex/Em 549/574 nm) was added to the cells, followed by incubation for 30 min at room temperature, protected from light. The FL-2 (585/42) channel was used to collect the fluorescence signal. The data were analyzed using the FlowJo software v.10 (BD Biosciences, San Jose, CA, USA). 

#### 4.7.4. Detection of Caspase 3/7 Activity 

Caspase-3/7 activity was detected according to the manufacturer’s instructions. To identify whether SS13 activated effector caspases-3 and -7, we used a DEVD (Asp-Glu-Val-Asp) peptide substrate conjugated with DNA binding dye. During the apoptotic process, the activation of caspases-3/7 in cells resulted in the cleavage of the recognition sequence in the DEVD peptide, the release of the dye, translocation to the nucleus, binding of the dye to DNA and high fluorescence. CellEvent™ Caspase-3/7 Green Detection Reagent was added to each flow cytometry tube and they were incubated for 30 min at 37 °C in the dark. Five minutes before measurement, the cells were stained with SYTOX™ AADvanced™ Dead Cell Stain. The FL-1 (530/30) vs. FL-4 (661/16) channels were used to collect fluorescence signals. The data were analyzed using the FlowJo software v.10 (BD Biosciences, San Jose, CA, USA).

#### 4.7.5. Analysis of Cytochrome *c* Release

Cytochrome c release was analyzed with flow cytometric analysis using the Cytochrome *c* Antibody (6H2) FITC Conjugate. The cell population was fixed with 4% paraformaldehyde for 15 min, washed with PBS and permeabilized with ice-cold 90% methanol. After 15 min incubation on ice, the cells were washed in PBS and stained with the conjugated antibody for 30 min at room temperature in the dark. Cells were washed in PBS again before analysis. The FL-1 (530/30) channel was used to collect the fluorescence signal. The data were analyzed with the FlowJo software v.10 (BD Biosciences, San Jose, CA, USA).

#### 4.7.6. Western Blot Analysis 

Western blot analysis was performed to evaluate protein levels. The HCT116 and Caco-2 cells were seeded in Petri dishes (1 × 10^6^ cells/dish) and incubated with SS13 for 24, 48 and 72 h. After treatment, Laemmle lysis buffer (glycerol, 20% sodium dodecyl sulfate (SDS), 1 M Tris/HCl (pH = 8.6), deionized H_2_O, protease and phosphatase inhibitors) and the process of sonication were used to prepare protein lysates. The colorimetric Pierce^®^ BCA protein assay kit (Thermo Scientific, Rockford, IL, USA) and the automated Cytation™ 3 Cell Imaging Multi-Mode Reader (Biotek, Winooski, VT, USA; wavelength 570 nm) were used to determine protein concentrations. The samples (concentration 40 ng) were loaded for separation by electrophoresis (100 V, 3 h) in SDS-PAA gel (10%), and then proteins from the gel were transferred to a polyvinylidene difluoride (PVDF) membrane using the iBlot^TM^ dry blotting system (Thermo Scientific, Rockford, IL, USA). The membranes were blocked in 5% dry non-fat milk (Cell Signaling Technology^®^, Danvers, MA, USA) or 5% bovine serum albumin (BSA; SERVA, Heidelberg, Germany) with TBS-Tween (pH = 7.4) for 1 h at room temperature to minimize nonspecific binding, and they were then incubated with primary antibodies (Table 4) overnight at 4 °C, followed by incubation with a horseradish peroxidase-conjugated secondary antibody. The expression of specific proteins was measured via an ECL chemiluminescent substrate (Thermo Scientific, Rockford, IL, USA) and detected with the MF-ChemiBIS 2.0 Imaging System (DNR Bio-Imaging Systems, Jerusalem, Israel) and iBright™ FL1500 Imaging System (Thermo Fisher Scientific, Cleveland, OH, USA). Analysis was performed densitometrically in the Image Studio Lite software (LI-COR Biosciences, Lincoln, NE, USA) and iBright Analysis software (Thermo Fisher Scientific, Cleveland, OH, USA). 

#### 4.7.7. RNA Isolation and Real-Time Polymerase Chain Reaction (RT-PCR)

The mRNA expression changes were analyzed by qRT-PCR. The total cellular RNA was isolated from HCT116 and Caco-2 cells with a Qiagen RNeasy^®^ Mini Kit (Qiagen, Manchester, UK), according to the manufacturer’s instructions. RNA quantification was performed spectrophotometrically using an Implen nanophotometer (Implen, Munich, Germany). Reverse transcription of tRNA (0.5 ng) into cDNA was performed using the RevertAid^TM^ H Minus First Strand c DNA synthesis kit (Fermentas GmbH, St. Leon-Rot, Germany). Subsequently, cDNA samples were used for quantitative RT-PCR in a 7500 FAST Real-Time PCR System (Applied Biosystems, Foster City, CA, USA). RT-PCR was carried out on an ABI 7500 RT-PCR device with a fast system using a Luna^®^ Universal qPCR Master Mix (BioLabs, New England Biolabs, Ipswich, MA, USA). Changes in the expression of genes involved in apoptosis, *CASP8*, *TNF*, *FASLG*, *BCL2*, *BCL2L1*, *BAX*, *BIK*, *BIRC5*, *XIAP* and *APAF1* were analyzed. A list of the used primer sequences is given in Table 5. The cycling parameters were as follows: 10 min of initial denaturation at 95 °C followed by 40 cycles of denaturation at 95 °C for 15 s, primer annealing at 60 °C (target gene) for 30 s and extension at 72 °C for 30 s (primer sequences were obtained from the RTPrimerDB database [54] and are listed in Table 5; supplied and validated by Sigma-Aldrich Chemie, Steinheim, Germany). Data were analyzed using the comparative threshold cycle (ΔΔCt) method and normalized to the expression level of the reference gene *GAPDH* in each sample. To ensure the purity of the amplification product, melting curves for each PCR reaction were generated. 

### 4.8. Statistical Analysis

Data are expressed as the mean ± standard deviation (SD) from three independent experiments. One-way analysis of variance (ANOVA), followed by the Bonferroni multiple comparisons test, was used for statistical evaluation. For the statistical analysis, * indicates *p* < 0.05, ** *p* < 0.01, *** *p* < 0.001 vs. untreated control. All experiments were performed in triplicate.

## 5. Conclusions

The presented data support the antiproliferative and pro-apoptotic potential of newly synthesized pyrrolidine derivative SS13 against colorectal cancer cells. Anti-proliferative assays revealed the higher sensitivity of Caco-2 cells to tested pyrrolidine derivative SS13, with IC_50_ values of 3.2 ± 0.1 μmol/L in the MTT assay and 1.59 ± 0.72 μmol/L in the BrdU proliferation assay. The IC_50_ values obtained in HCT116 cells were 3.2 ± 0.1 μmol/L in the MTT assay and 6.46 ± 2.84 μmol/L in the BrdU proliferation assay. The tested compound activated both the intrinsic and extrinsic apoptotic pathways. SS13-induced apoptosis was linked to changes in MMP, the release of cytochrome *c*, the increased activity of caspase-3/7, PARP cleavage and the modulation of Bcl-2 family proteins. *FASLG* and *TNF* overexpression, as well as caspase-8 cleavage, contributed to the extrinsic apoptotic pathways. Moreover, pyrrolidine derivative SS13 suppressed the migration of colorectal cancer cells. Although further studies should be performed to further elucidate the antiproliferative effects of SS13, our results suggest that pyrrolidine derivative SS13 may have potential in the prevention and treatment of colorectal cancer.

## Figures and Tables

**Figure 1 ijms-24-11696-f001:**
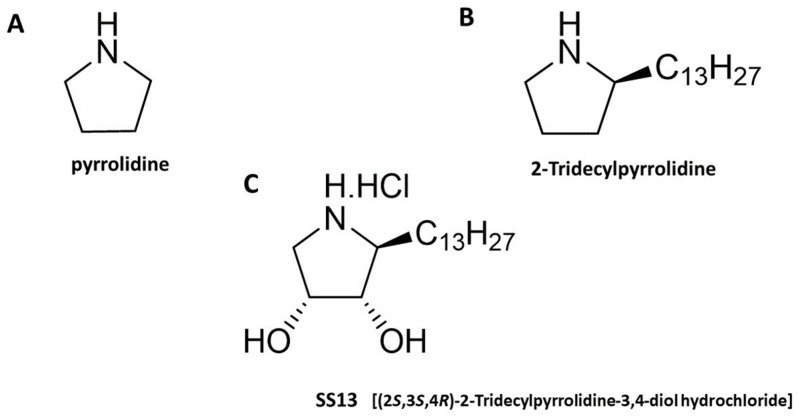
Chemical structures of pyrrolidine (**A**), 2-tridecylpyrrolidine (**B**) and SS13 [(2*S*,3*S*,4*R*)-2-tridecylpyrrolidine-3,4-diol hydrochloride] (**C**). The title compounds are named according to the rules of the International Union of Pure and Applied Chemistry (IUPAC) nomenclature.

**Figure 2 ijms-24-11696-f002:**
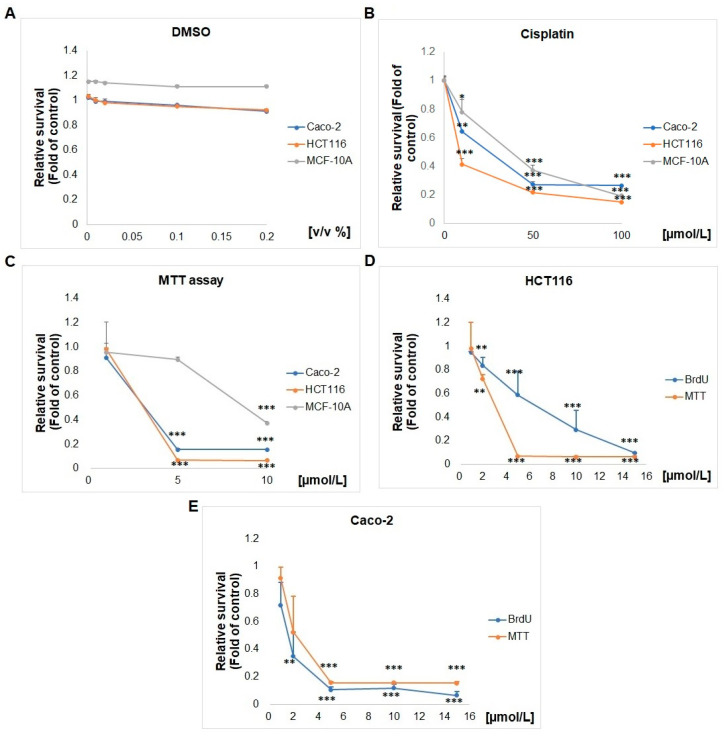
Effect of DMSO (**A**), cisplatin (**B**) and SS13 (**C**–**E**) on the proliferation of HCT116, Caco-2 and MCF-10A cells. DMSO equivalents: 1 µM SS13 = 0.002 *v*/*v* % of DMSO; 5 µM SS13 = 0.01 *v*/*v* % of DMSO; 10 µM SS13 = 0.02 *v*/*v* % of DMSO; 50 µM SS13 = 0.1 *v*/*v* % of DMSO; 100 µM SS13 = 0.2 *v*/*v* % of DMSO. The presented data show the mean values ± standard deviation across three independent experiments. Statistical significance: * *p* < 0.05, ** *p* < 0.01, *** *p* < 0.001 vs. untreated cells (control).

**Figure 3 ijms-24-11696-f003:**
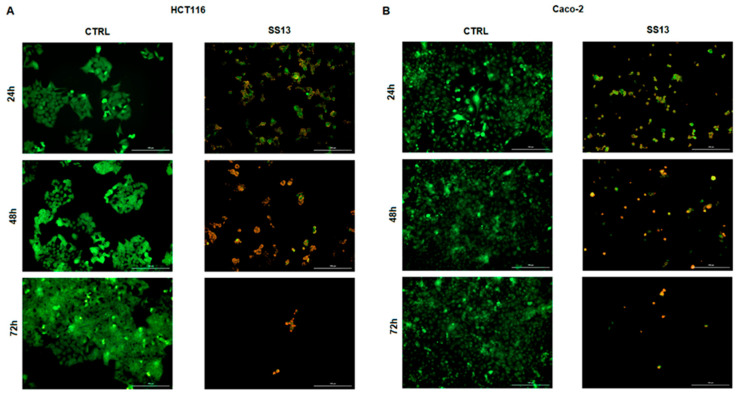
Fluorescence microscopic analysis of SS13-induced apoptosis in HCT116 (**A**) and Caco-2 (**B**) cell lines after 24, 48 and 72 h of treatment (7 µM for HCT116, 2.5 µM for Caco-2) using AO/PI staining. Green indicates live cells, yellow indicates cells in the early stage of apoptosis, orange indicates cells in the late stage of apoptosis and red represents dead/necrotic cells. Representative figure of three independent experiments. Magnification: 100×.

**Figure 4 ijms-24-11696-f004:**
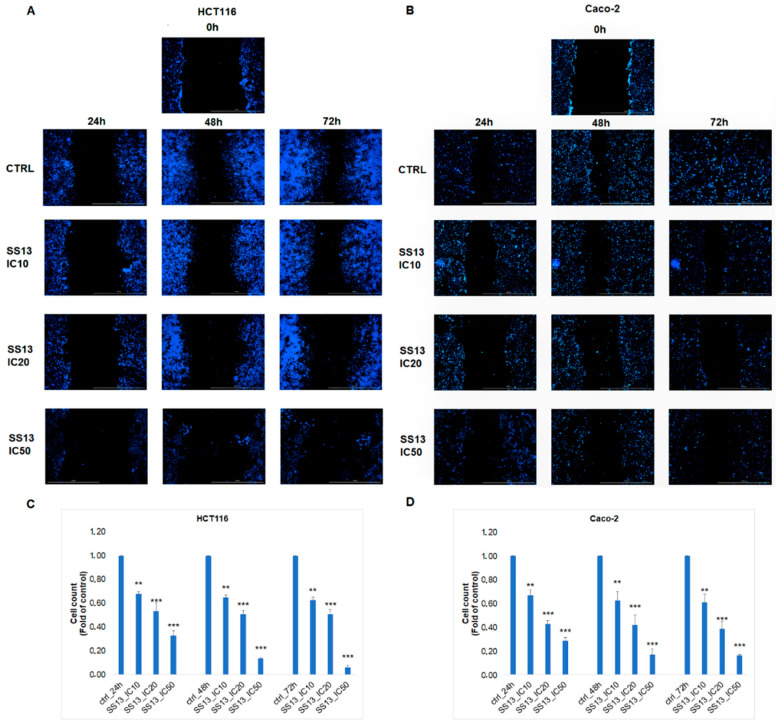
Scratch assay was performed on HCT116 (**A**) and Caco-C (**B**) cells treated with IC_10_–IC_50_ doses of SS13 at three different times and they were compared to untreated groups. The figures depict three independent experiments. The graph represents the number of HCT116 (**C**) and Caco-2 (**D**) cells in (**A**) and (**B**), respectively. Values represent the means ± standard deviations of three independent experiments. Statistical significance: ** *p* < 0.01, *** *p* < 0.001 vs. untreated cells (control).

**Figure 5 ijms-24-11696-f005:**
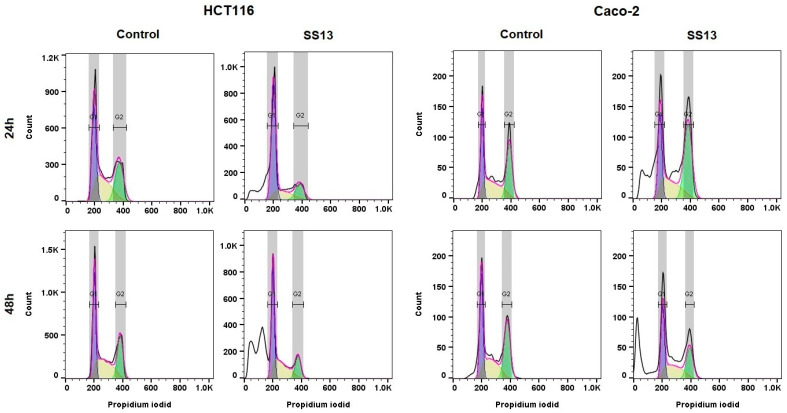
Representative histograms of flow cytometry cell cycle analyses of HCT116 and Caco-2 cell lines after 24 and 48 h of treatment with SS13 (7 µM for HCT116, 2.5 µM for Caco-2).

**Figure 6 ijms-24-11696-f006:**
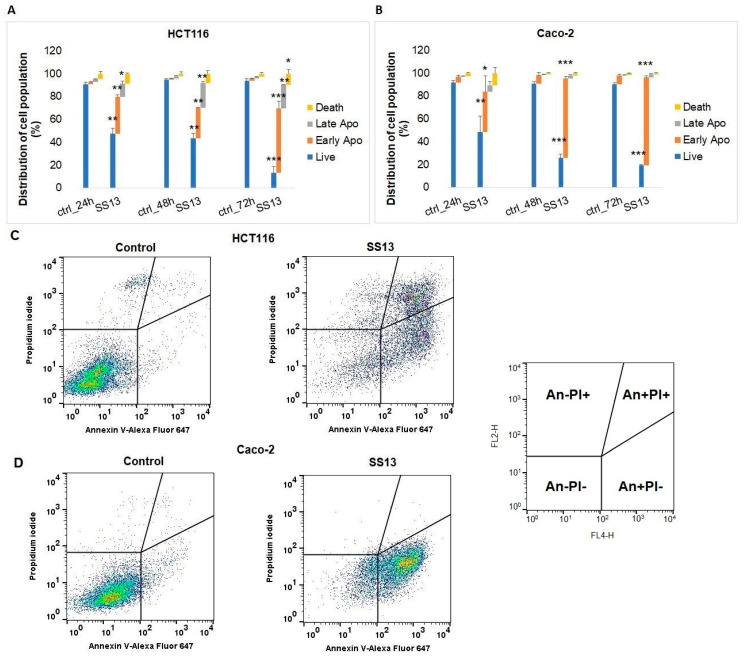
Annexin V/PI analysis of the incidence of apoptosis in SS13-treated HCT116 (7 µM) and Caco-2 (2.5 µM) cell populations (**A**,**B**). Representative Annexin V/PI dot plot of HCT116 (**C**) and Caco-2 (**D**) cells after 72 h of incubation with tested compound SS13. The presented data are the means ± standard deviations of three independent experiments. Statistical significance: * *p* < 0.05, ** *p* < 0.01, *** *p* < 0.001 vs. untreated cells (control).

**Figure 7 ijms-24-11696-f007:**
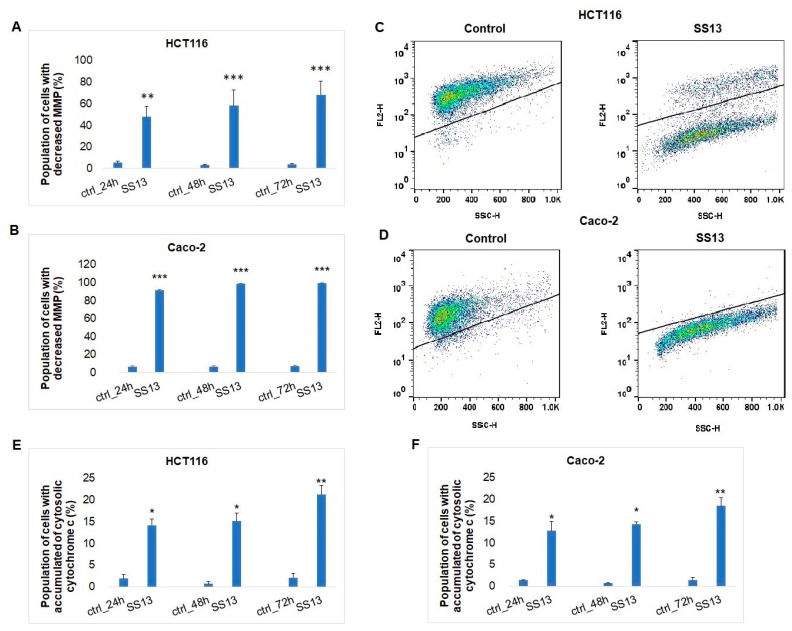
Effect of SS13 (7 µM for HCT116 and 2.5 µM for Caco-2) on MMP and cytochrome c release. Analyses of MMP changes in HCT116 (**A**) and Caco-2 (**B**) after 24, 48 and 72 h of treatment with SS13. Representative dot plot of MMP after 72 h of incubation with SS13 in HCT116 (**C**) and Caco-2 (**D**). Relative release of cytochrome c in HCT116 (**E**) and Caco-2 (**F**) cells. The data show the means ± standard deviations of three independent experiments. Statistical significance: * *p* < 0.05, ** *p* < 0.01, *** *p* < 0.001 vs. untreated cells (control).

**Figure 8 ijms-24-11696-f008:**
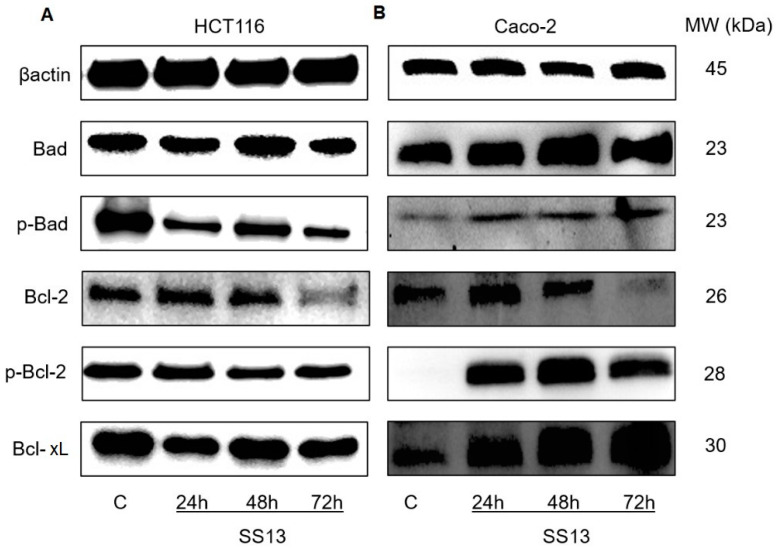
Western blot analysis of apoptosis-related proteins in HCT116 (**A**) and Caco-2 (**B**) cells after treatment with SS13 after 24, 48 and 72 h of incubation (7 µM for HCT116, 2.5 µM for Caco-2).

**Figure 9 ijms-24-11696-f009:**
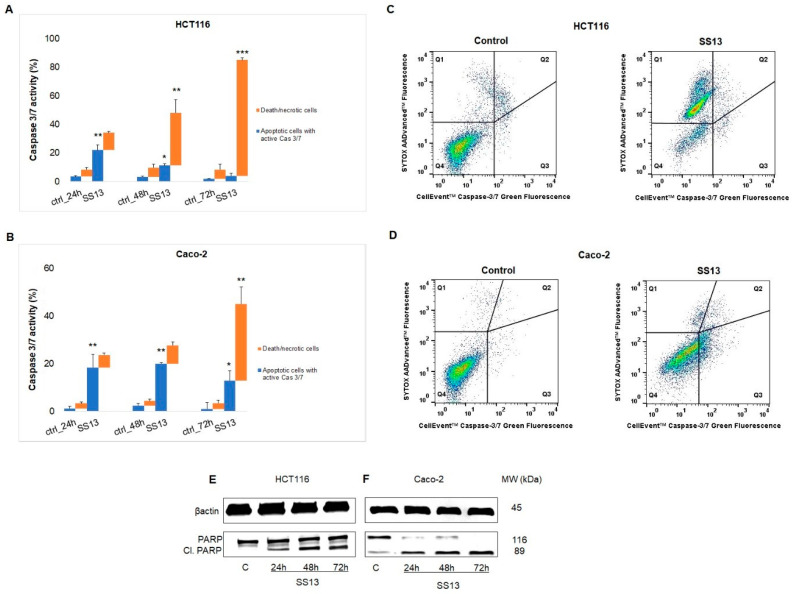
Effect of SS13 (7 µM for HCT116, 2.5 µM for Caco-2) on caspase-3/7 activity and PARP cleavage. Flow cytometric analysis of activated caspase-3/7 after 24, 48 and 72 h of SS13 treatment in HCT116 (**A**) and Caco-2 (**B**) cells. Representative dot plot of caspase-3/7 activation after 72 h incubation with SS13 in HCT116 (**C**) and Caco-2 (**D**), where quadrant Q3 represents apoptotic cell population with active caspase-3/7; Q1 + Q2 dead/necrotic cells; Q4 live cells. Western blot analysis of PARP after 24, 48 and 72 h of treatment with the tested compound in HCT116 (**E**) and Caco-2 (**F**) cells. The values shown are the means of three independent experiments compared to the control. Statistical significance: * *p* < 0.05, ** *p* < 0.01, *** *p* < 0.001 vs. untreated cells (control).

**Figure 10 ijms-24-11696-f010:**
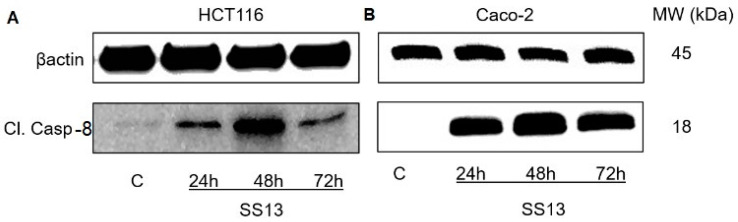
Western blot analysis of cleaved caspase-8 after 24, 48 and 72 h of SS13 treatment in HCT116 ((**A**) 7 µM) and Caco-2 ((**B**) 2.5 µM) cells.

**Table 1 ijms-24-11696-t001:** The IC_50_ values of HCT116, Caco-2 and MCF-10A treated with SS13 and cisplatin (Cis-Pt) and their selectivity indexes.

	Colon Cancer Cell Lines		Normal Epithelial Cell Line
	HCT116	Caco-2	MCF-10A
SS13	3.2 ± 0.1	2.2 ± 1.5	8.8 ± 0.1
Cis-Pt	9.0 ± 0.4	30.7 ± 0.8	25.9 ± 0.3
Selectivity index for SS13	2.75	4	1
Selectivity index for Cis-Pt	2.88	0.84	1

**Table 2 ijms-24-11696-t002:** Effect of pyrrolidine SS13 on cell cycle (average data).

HCT116	Time	Sub-G0/G1	G1	S	G2/M
ctrl	24 h	0.5 ± 0.4	35.1 ± 2.2	34.2 ± 5.0	30.3 ± 3.1
SS13		14.1 ± 4.0 *	46.1 ± 4.6 *	21.9 ± 4.7 *	18.0 ± 4.0 *
ctrl	48 h	1.2 ± 0.5	35.2 ± 1.2	39.7 ± 3.7	24.0 ± 3.5
SS13		41.4 ± 3.6 **	24.4 ± 4.2 *	21.1 ± 1.6 *	13.1 ± 2.8 *
ctrl	72 h	0.9 ± 0.4	57.6 ± 3.2	27.4 ± 3.3	14.1 ± 3.5
SS13		19.3 ± 3.1 *	40.6 ± 2.8 *	24.6 ± 3.4	15.5 ± 2.3
**Caco-2**					
ctrl	24 h	1.3 ± 0.4	33.2 ± 2.7	31.0 ± 2.1	34.4 ± 3.2
SS13		21.7 ± 8.6 *	28.3 ± 5.2 *	19.1 ± 1.9 *	30.9 ± 3.5
ctrl	48 h	2.4 ± 1.4	33.4 ± 1.8	28.0 ± 2.3	36.2 ± 2.0
SS13		32.3 ± 5.3 **	28.0 ± 1.4 *	22.0 ± 3.3 *	17.7 ± 4.2
ctrl	72 h	0.7 ± 0.5	35.5 ± 1.8	28.9 ± 1.7	34.9 ± 1.9
SS13		24.2 ± 7.1 **	29.2 ± 2.2 *	22.5 ± 1.1 *	24.1 ± 3.9 *

Statistical significance: *n* = 3; * *p* < 0.05, ** *p* < 0.01 vs. untreated cells (control).

**Table 3 ijms-24-11696-t003:** The effect of pyrrolidine SS13 on gene expression in HCT116 and Caco-2 cells.

Pro-Apoptotic				Anti-Apoptotic			
Caco-2							
Gene	24 h	48 h	72 h	Gene	24 h	48 h	72 h
** *BAX* **	2.514 ± 0.3 **	2.100 ± 0.2 **	5.236 ± 0.3 ***	** *BCL2* **	0.891 ± 0.1	2.234 ± 0.2 **	6.251 ± 0.6 ***
** *BIK* **	2.863 ± 0.1 **	2.207 ± 0.2 **	0.393 ± 0.1 ***	** *BCL2L1* **	2.107 ± 0.1 **	3.775 ± 0.6 ***	1.240 ± 0.7
** *APAF1* **	0.565 ± 0.3 *	3.668 ± 0.6 ***	1.535 ± 0.4 *	** *XIAP* **	1.260 ± 0.4	2.281 ± 0.2 **	0.402 ± 0.1 ***
** *CASP8* **	1.408 ± 0.01 *	0.778 ± 0.1	0.441 ± 0.1 **	** *BIRC5* **	1.905 ± 0.3 *	1.849 ± 0.4 *	1.049 ± 0.2
** *TNF* **	1.383 ± 0.3	2.136 ± 0.1 **	3.566 ± 0.4 ***				
** *FASLG* **	1.265 ± 0.2	7.075 ± 0.5 ***	0.704 ± 0.1*				
**HCT116**							
** *BAX* **	1.831 ± 0.3 *	6.548 ± 1.4 ***	1.968 ± 0.2 *	** *BCL2* **	0.353 ± 0.2 ***	0.281 ± 0.1 ***	0.179 ± 0.4 ***
** *BIK* **	0.512 ± 0.1 **	0.329 ± 0.1 ***	3.241 ± 0.2 ***	** *BCL2L1* **	0.078 ± 0.3 ***	0.117 ± 0.2 ***	0.556 ± 0.2 **
** *APAF1* **	0.742 ± 0.2*	0.603 ± 0.1 **	3.564 ± 0.4 ***	** *XIAP* **	0.313 ± 0.2 ***	0.395 ± 0.2 ***	0.409 ± 0.1 ***
** *CASP8* **	2.312 ± 0.3 **	1.496 ± 0.1 *	1.021 ± 0.7	** *BIRC5* **	2.301 ± 0.3 **	4.637 ± 0.4 ***	10.460 ± 1.6 ***
** *TNF* **	1.800 ± 0.2 *	1.027 ± 0.4	0.646 ± 0.1 **				
** *FASLG* **	1.419 ± 0.1 *	1.636 ± 0.1 *	1.724 ± 0.3 *				

Data were analyzed using the ΔΔCt comparison method and normalized to the expression level of the reference gene GADPH in each sample. Statistical significance: * *p* < 0.05, ** *p* < 0.01, *** *p* < 0.001 vs. GADPH expression.

**Table 4 ijms-24-11696-t004:** List of Western blot antibodies.

Primary Antibody	Mr (kDa)	Origin	Dilution	Catalogue No.	Manufacturer
Bad (D24A9) Rabbit mAb	23	Rabbit	1:1000	#9239	Cell Signaling Technology^®^, Danvers, MA, USA
Phospho-Bad (Ser136) Antibody	23	Rabbit	1:1000	#9295	
Phospho-Bcl-2 (Ser70) (5H2) Rabbit mAb	28	Rabbit	1:1000	#2827	
Bcl-xL (54H6) Rabbit mAb	30	Rabbit	1:1000	#2764	
Cleaved Caspase 8 (Asp374) (18C8) Rabbit mAb	43/41/18	Rabbit	1:1000	#9496	
PARP (46D11) Rabbit mAb	116/89	Rabbit	1:1000	#9532	
β-Actin (8H10D10) Mouse mAb	45	Mouse	1:1000	#3700	
Anti-Bcl2 Antibody [100/D5]	26	Mouse	1:500	ab692	Abcam
**Secondary Antibody**	**Mr** (**kDa**)	**Origin**	**Dilution**	**Catalogue No.**	**Manufacturer**
Anti-Rabbit IgG HRP-Linked Antibody	-	Goat	1:1000	#7074	Cell Signaling Technology, Danvers, MA, USA
Anti-Mouse IgG HRP-Linked Antibody	-	Goat	1:1000	#7076	

**Table 5 ijms-24-11696-t005:** Primer sequences used in real-time PCR analysis.

Gene	Forward Primer	Reverse Primer
** *GAPDH* **	5′-TGCACCACCAACTGCTTAGC-3′	5′-GGCATGGACTGTGGTCATGAG-3′
** *BAX* **	5′-CCCGAGAGGTCTTTTTCCGAG-3′	5′-CCAGCCCATGATGGTTCTGAT-3′
** *BCL2* **	5′-TCCATGTCTTTGGACAACCA-3′	5′-CCAGCCCATGATGGTTCTGAT-3′
** *BCL2L1* **	5′-TTACCTGAATGACCACCTA-3′	5′-ATTTCCGACTGAAGAGTGA-3′
** *CASP8* **	5′-AGAGTCTGTGCCCAAATCAAC-3′	5′-GCTGCTTCTCTCTTTGCTGAA-3′
** *FASLG* **	5′-CACTTTGGGATTCTTTCCAT-3′	5′-GTGAGTTGAGGAGCTACAGA-3′
** *TNF* **	5′-GGCGTGGAGCTGAGAGATAAC-3′	5′-GGTGTGGGTGAGGAGCACAT-3′
** *BIK* **	5′-TCTGCAATTGTCACCGGTTA-3′	5′-TTGAGCACACCTGCTCCTC-3′
** *XIAP* **	5′-AGTGGTAGTCCTGTTTCAGCATCA-3′	5′-CCGCACGGTATCTCCTTCA-3′
** *BIRC5 (survivin)* **	5′-TGCCTGGCAGCCCTTTC-3′	5′-CCTCCAAGAAGGGCCAGTTC-3′
** *APAF1* **	5′-TCCATGTATGGTGACCCATCC-3′	5′-AAGGTGGAGTACCACAGAGG-3′

## Data Availability

Data are available upon request from the corresponding author.

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
