# Peer review of "Involvement of Both Extrinsic and Intrinsic Apoptotic Pathways in Tridecylpyrrolidine-Diol Derivative-Induced Apoptosis In Vitro"

_ijms, 2023, doi:10.3390/ijms241411696_

Round 1
Reviewer 1 Report
Authors of the publication titled “Involvement of both extrinsic and intrinsic apoptotic pathways 2 in tridecylpyrrolidine-induced apoptosis in vitro” have investigated the antiproliferative and pro-apoptotic effect of (2S,3S,4R)-2-tridecylpyrrolidine-3,4-diol hydrochloride on colon cancer cells.
I recommended accepting after major corrections.
1- The introduction is short, the authors need to add a part about extrinsic and intrinsic apoptotic pathways.
I suggest reading this publication.
https://bmccancer.biomedcentral.com/articles/10.1186/s12885-021-08167-y
2- Line 44 and 45 (Chemically, pyrrolidines are tetrahydro pyrroles or cyclic amines whose five-membered ring consists of four carbon atoms and one nitrogen atom)
Authors need to represent the structure of tetrahydro pyrroles and not describe it in the text.
3- Results:
3.1. Authors need to use control in antiproliferative assays, DMSO is a negative control, and authors need to use a positive control which is approved as anti-cancer medication.
3.2. lines 105-107 explain the mechanism of AO/PI fluorescent staining.
(AO can diffuse into cells with 105 intact membrane integrity and PI to cells with a disrupted membrane. Due to this method, cell populations can be divided into living (green), apoptotic (yellow/orange), and dead (red) cells).
This part should move to the introduction and not the Results.
4- Conclusion:
The authors should add the data of antiproliferative assays to the conclusion.
5- Reference:
I suggest the authors update the references with recent ones, there have only been around 4 references in the last 3 years and most of the others are old.
Minor corrections are required in English.
Line 16 (we investigated antiproliferative), the correct one is we investigated the antiproliferative.
Author Response
Please sea attachment

Reviewer 2 Report
This research investigates the effects of a new dihydroxypyrrolidine derivative on two colon cancer cell lines, examining their phenotypic and molecular effects related to cell proliferation, migration, and cell death. The study is important as it helps fill gaps in our knowledge about the biological activities of compounds containing a five-membered pyrrolidine ring, considered a versatile scaffold for developing new biologically active compounds.
The attached pdf file (33 items) provides specific suggestions and comments for authors.

The manuscript needs minor language editing.
Author Response
Please sea attachment

Reviewer 3 Report
This manuscript by Nosalova et al presents a study on the evaluation of a newly synthesized compound called SS13 and its involvement in both extrinsic and intrinsic apoptotic pathways in vitro. I find this work to be interesting and well-written, and I believe it has the potential to be published in IJMS. The experimental data provided strongly supports the authors' conclusions.
However, I have a few suggestions for minor adjustments that could improve the manuscript:
1. The title appears to be somewhat confusing. It is unclear whether tridecylpyrrolidine or SS13 is inducing apoptosis. Furthermore, it would be helpful to clarify the specific type of tridecylpyrrolidine being referred to, such as whether it is N-substituted or not. I recommend that the authors rephrase the title to better describe the focus of their work and address these concerns.
2. When referring to the compounds, it would be more accurate to use the term "pyrrolidine derivatives" instead of just "pyrrolidine." This change will help to specify that SS13 is a derivative of pyrrolidine.
3. It would be beneficial to move the figure of SS13 earlier in the document, preferably by the end of the introduction. By doing so, readers will have a visual reference for SS13 while reading the subsequent sections, aiding in their understanding of the research.
4. Lastly, and most importantly, I believe that the authors should provide additional information about SS13. It would be helpful to explain what SS13 is, the rationale behind designing this compound, and why the authors chose to work with it. Currently, the manuscript only briefly mentions in line 397 that Pyrrolidine SS13 was selected based on screening from a series of tested substances. It would be beneficial if the authors could elaborate on the specific data or criteria that led them to choose SS13 for further investigation.
Quality of English Language is fine
Author Response
Please sea attachment

Round 2
Reviewer 1 Report
Authors carried out the required corrections. My comment accept the manuscript in present form.
Author Response
Dear reviewer, we appreciate your effort.
Best regards
Reviewer 2 Report
Most comments of this reviewer have been addressed. Changes implemented in the revised version considerably improved the manuscript.
The remaining issues are only related to the cell cycle analysis by flow cytometry. That section still needs to be improved to address inadequate reporting and interpretation deficiency.
The reviewer thinks that this section is not essential for the manuscript, which even without this section provides sufficient evidence on the induction and nature of apoptosis in cells treated with investigated pyrrolidine derivative. On the other hand, the conclusion derived from this section about the non-induction of cell cycle arrest is problematic and unsupported.
---------------------------
Comments relevant only to cell cycle analysis by flow cytometry
1. This reviewer's original comments (first peer-review):
---To properly determine cell cycle phases, the cell cycle analysis data need to be re-analyzed. Authors need to produce a separate histogram without sub-G fraction and use an appropriate model for histogram deconvolution.
---The assay is suitable for detecting apoptosis but not for quantifying apoptosis together with the distribution of cells to all cell cycle phases. Authors can determine cell distribution after the sub-G1 population is gated out. Then a new histogram can be fit, and the distribution of cells in cell cycle phases can be properly determined.
2. The response by the authors needs to be more satisfactory. This reviewer's rebuttal is shown in bold in the author's response text below:
--- "We do not agree in several points you mentioned. As we statemented in manuscript and you pointed we analysed population of cells (untreated and treated) to observe cell cycle changes include occurrence of subpopulation of cells with fragmented DNA (sub G0/G1).
We and many other researchers (see links) not agree to distract subG0 from analyses and re-analyse."
This is an appeal to authority insisting that a claim is valid because a good authority said it was true without any other supporting evidence.
---- "The population of events (signals) must be taken together."
Why would it be helpful to include signals that may not be uniquely attributable to original cells (in a 1:1 relationship)? There can be many fragments from any single cell nucleus that constitute sub-G0/G1 population.
--- "Flow cytometry methods is based on DNA staining by intercalating propidium iodid. So de facto, no cells but nuclei or better nuclear DNA are counting based fluorescence signals. Yes, if nuclei are fragmented, there are more fragments and in analyses they are clearly localised before G1 peak. But the results is not cell count or better event count but the results is % of events in 4 distinguished subpopulations."
This needs to be corrected. You calculate the "% of cells in sub-G1" from the number of events registered by fluorescence, and these events, which may be nuclear fragments, may not correspond to original cells in a 1:1 relationship.
--- "Distracting subG0 subpopulation from complex cell cycle analyses did not change the pattern of G1, S, G2/M distribution. The peaks will be only higher depends how big subpopulation where distracted. Your advice technically have only little benefits in terms that only 1 think will change. Only % of G1, S, G2/M subpopulations increase evenly according the % of subG0 subpopulations but pattern remain."
This is an incorrect statement, as demonstrated below:
I have determined quantitative values for cell cycle distributions using a ruler and the size of the bars you provided in Figure 4. Please, note that it is unfortunate that the aggregated statistics for cell cycle analysis (mean % values, SDs from replications) have not been reported.
Contingent on the bar graph accuracy, this reviewer found the following changes in cell cycle distributions for 48-h exposures:
HCT116 cells:
-G0/G1: 33.3% (control); 68% (treated)
-S: 31.5% (control); 16% (treated)
-G2/M: 35.2% (control); 16% (treated)
CACO-2
-G0/G1: 41.2% (control); 53.8% (treated)
--S: 17.6% (control); 15.4% (treated)
--G2/M: 41.2% (control); 30.7% (treated)
So, removing sub-G0/G1 particles from the analysis and limiting the analysis to alive cells has considerably changed the cell cycle distribution pattern. Moreover, if the Figure 4 bars are correct, the authors' conclusion that the tested compound does not induce cell cycle arrest is incorrect. Both HCT116 and CACO-2 cells show a substantial treatment-induced relative increase in G0/G1 cells indicating cell cycle arrest, consistent with the conclusions of Omar et al. (2017) that is cited in the manuscript as reference #33. It is also consistent with your findings reported in this manuscript that showed inhibition of DNA proliferation by the BrdU assay.
--- "Therefore, we and many others researchers not doing that.
https://doi.org/10.1080/01635581.2019.1616780
https://doi.org/10.1155/2020/5924856
https://doi.org/10.1007/s10495-019-01539-7
https://doi.org/10.1002/jcp.28552
https://www.sciencedirect.com/science/article/pii/S0378874122008881"
It should be noted that current published biomedical research has a considerable proportion of irreproducible results ("reproducibility crisis") and that not all published reports provide high strength of evidence. As a result, simply stating that others do the same is generally not a sufficient argument that would resolve any identified methodological issues. If authors want to use this type of argument, they would need to select much stronger sources, such as consensus methodological papers, peer-reviewed protocols, or guidelines published by expert groups with robust peer-review, such as OECD and others.
Also note that according to the author's conclusion stated on pages 6 and 12 of the manuscript, "SS13 did not induce any cell cycle arrest". However, this conclusion that questions findings reported by other investigators, such as Omar et al. (2017) (cited as reference #33) is inadequately supported by this cell cycle analysis.
Let us consider for a moment a scenario in which an untreated cell culture has the following distribution: G0/G1=60%, S=20%, G2/M=20%.
Let us assume for simplicity that we started with 10,000 cells and treated it with a compound that induced full cell cycle arrest at G0/G1 phase (6,000 cells) and killed 50% of cells with S phase (originally 2,000 cells, the remaining 1,000 cells) and 50% cells in G2/M phase (originally 2,000 cells, the remaining 1,000 cells). If we determined cell cycle distribution that would include 2,000 apoptotic cells using the authors' reasoning, we would have 2,000 apoptotic cells (20%), 6,000 G0/G1 cells (60%), 1,000 S cells (10%) and 1,000 G2/M cells (10%). This approach would find apoptosis but it would fail to detect cell cycle arrest in the G0/G1 population.
However, if sub-G0/G1 cells are removed from this consideration, we would find 6,000 G0/G1 cells out of all 8,000 considerable cells, which gives 75% G0/G1 cells and clear evidence of cell cycle arrest.
This reviewer finds that the problem described above is a conceptual problem in authors' evaluation of cell cycle data when assessing cell chemically-induced cycle-arresting effects in the presence of sub-G0/G1 populations.
In addition, there are technical issues that the authors need to address:
1. Description of this analysis in the method section is still limited. For an expert community, it is less interesting to read how this assay works. However, it is essential to understand what the authors have done to generate these results. There are many possible variations in the execution of this experiment, and one needs to provide enough information so that others can perform a replication of this study if they are so inclined.
- After my first request, the authors indicated how they removed signals of clusters of cells from the data and which SW they have used. Of note, the SW version and supplier are also important to provide to avoid confusion.
- Authors still need to indicate which method (algorithm) has been used for fitting experimental data to generate histograms and deconvolute peaks to G0/G1, S, and G2/M phases. This is an important information rather than a formality. Note that FloJo offers two models to fit the data: Watson Pragmatic and Dean-Jett-Fox. The two models differ in their mathematical calculations of each phase of the cell cycle. Consequently, results from one model may vary quite significantly from the other. It is good laboratory practice to consistently use the same model throughout a study when reporting or publishing statistics, and the authors need to indicate which model they have used.
- The authors did not provide RMSD metrics that would show the goodness of the model fitting for cell cycle analysis. Without it, the readers can only take the authors' word that this was adequately fit.
- The authors did not show the actual representative DNA histograms, which would allow readers to see if there were aneuploid cell populations that could impact on the interpretation of results.
- The authors did not indicate whether they used any QC particles to verify flow cytometry performance (linearity)
- The authors need to provide a summary descriptive statistics for % populations in individual phases of the cell cycle and SDs from replicate experiments
Based on the review described above, this reviewer disagrees with the following statements, which have to be reconsidered.
-------(P6) 2.4 Cell cycle analysis Flow cytometric analysis was performed to identify whether the tested compound induced cell cycle arrest. We noticed a reduction in the number of cells in the G1 phase after 72h incubation with SS13 in both cell lines (Error! Reference source not found.5). In addition, a significant accumulation of cells in the sub-G0 population was observed, which is a typical marker of apoptosis. These results showed no cell cycle arrest after SS13 treatment, but we determined time-dependent apoptosis in both HCT116 and Caco-2 cells.
------(P12): Some studies have shown that G1/S or G2/M phase cell cycle arrest can mediate pyrrolidine derivates' cytotoxic effect [18,33]. Flow cytometric analysis was used to determine whether SS13 induced changes in the cell cycle. Our results showed that the tested pyrrolidine derivative did not induce any cell cycle arrest, as described in previous.
Author Response
Dear reviewer,
we want to thank you for erudated explanation of cell cycle analyses. We re-analysed data based on your knowledge using Dean-Jett-Fox model for all samples. Now we can see G1 arrest you mentioned. New table with averaged data and new figure with representative histogram were added to text. All relevant section were upgraded based new data.
Best regards